# Performance Evaluation of BD Phoenix and MicroScan WalkAway Plus for Determination of Fosfomycin Susceptibility in *Enterobacterales*

**DOI:** 10.3390/antibiotics12071106

**Published:** 2023-06-26

**Authors:** Alessandro Bondi, Antonio Curtoni, Marco Peradotto, Elisa Zanotto, Matteo Boattini, Gabriele Bianco, Marco Iannaccone, Anna Maria Barbui, Rossana Cavallo, Cristina Costa

**Affiliations:** 1Microbiology and Virology Unit, University Hospital Città della Salute e della Scienza di Torino, University of Turin, 10126 Turin, Italy; alessandro.bondi@unito.it (A.B.);; 2Department of Public Healt and Pediatric Sciences, University of Turin, 10126 Turin, Italy; 3Clinical Laboratory, Microbiology Unit, Sant’Andrea Hospital, 13100 Vercelli, Italy

**Keywords:** fosfomycin, agar dilution, microdilution method, *Enterobacterales*

## Abstract

Background: Fosfomycin is an old bactericidal drug that has gained increasing interest in the last decade for its potential use in multi-drug resistant gram-negative infections. However, evidence on fosfomycin susceptibility testing reports a poor correlation between commercial methods vs. reference agar dilution (AD) for *Enterobacterales* (EB). The study aimed at assessing the performance of two automated systems for the determination of fosfomycin susceptibility in *EB* clinical isolates. Methods: Fosfomycin susceptibility testing results of two collections of 100 non-duplicate clinical EB strains obtained using two different platforms (BD Phoenix and MicroScan WalkAway Plus) were compared with those obtained by AD. Categorical agreement (CA), major error (ME) and very major error (VME) rates were calculated. Results: BD Phoenix exhibited a 6.9% rate of false-resistant results and achieved a CA of 69%, whereas MicroScan WalkAway Plus achieved 3.7% of false-resistant results and 72% of CA. Both automated systems showed poor detection of resistant isolates, with 49.1% and 56.2% of false-susceptible results for BD Phoenix and Microscan WalkAway Plus, respectively. Conclusions: Overall, agar dilution remains the most suitable method for routine laboratory antimicrobial susceptibility testing of fosfomycin on *Enterobacterales* strains, given the poor performance of automated systems. The application of both automated systems, in the clinical laboratories reporting of fosfomycin, should be reviewed in light of the accuracy results falling below the acceptable threshold.

## 1. Introduction

The emerging and global spread of extended-spectrum β-lactamase- and carbapenemase-producing *Enterobacterales* (ESBLp-EB and Cp-EB) has reduced the number of effective drugs, and new therapeutic options are highly desirable [1,2]. Fosfomycin is an old bactericidal drug that inhibits peptidoglycan synthesis, interfering with the formation of citoplasmatic precursor UDP *N*-UDP *N*-acetylmuramic acid (UDP-MurNAc) [3]. In the last years, its use in clinical practice has been a subject matter of increasing interest. Given its broad-spectrum activity, ability to penetrate into biofilm, pharmacokinetic profile, and safety, it is currently recommended for the treatment of lower urinary tract infections [3,4,5]. Additionally, it serves as a reliable option for infections caused by multi-drug (MDR) resistant gram-negative bacilli and carbapenem-sparing therapy [5,6,7,8,9,10,11]. The European Committee for Antimicrobial Susceptibility Testing (EUCAST) provide two clinical breakpoints (BPs) based on administration route (*iv* and *os* administration) for *Enterobacterales* spp. The *os* BPs are valid only for uncomplicated urinary tract infection caused by *E. coli* [12].

According to international guidelines [12,13], agar dilution (AD) represents the reference method for fosfomycin susceptibility testing. However, due to its time-consuming protocol, fosfomycin susceptibility testing is largely and routinely performed using automated systems. Recently, we investigated the performance of automatic systems against *Staphylococcus aureus* and found low categorical agreement [14]. In order to investigate the cause of this low reliability, whether it was attributable to specific bacterial specie or the method, we have pointed our attention on Gram-negative bacteria. Limited published evidence reports a poor correlation between commercial methods and reference AD for EB isolates [15,16,17,18]. The aim of this study was to assess the performance of two automated systems, BD Phoenix (Becton Dickinson, MD, USA) and MicroScan WalkAway Plus (Beckman Coulter, CA, USA), in comparison with reference AD for the determination of fosfomycin susceptibility in EB clinical isolates.

## 2. Results

Two different collection of EB were tested to evaluated the performance of BD Phoenix (*Klebsiella pneumoniae*
*n* = 50, *Escherichia coli*
*n* = 46, *Enterobacter cloacae*
*n* = 4) and MicroScan WalkAway (*E. coli*
*n* = 42, *K. pneumoniae*
*n* = 27, *E. cloacae*
*n* = 17, *Proteus mirabilis*
*n* = 10, *Morganella morganii*
*n* = 3, *Citrobacter koseri*
*n* = 3).

Phenotypic characterization of the EB tested using BD Phoenix and MicroScan WalkAway Plus is reported in Table 1. The two automated systems were used to assess similar rates of ESBLp-EB (33% vs. 30%) and Cp-EB (17% vs. 16%). A total of 17 Cp-EB strains were tested by BD Phoenix (KPC-producers *n* = 14; VIM-producing *E. cloacae n* = 2; OXA-48-producing *K. pneumoniae n* = 1). Similarly, 16 Cp-EB were tested by Microscan WalkAway Plus (KPC-producers *n* = 10; VIM-producers *n* = 5; OXA-48-producing *K. pneumoniae n* = 1). The fosfomycin resistance rate of the EB strains detected by BD Phoenix and MicroScan WalkAway Plus was 57% and 46%, respectively. The overall fosfomycin susceptibility obtained with the reference method was 48.6% (*n* = 97).

The comparison between fosfomycin susceptibility testing results obtained with the automated systems and the reference agar dilution method for overall EB isolates and according to the most frequent species is reported in Table 2 and Table 3.

Categorical agreement, major error and very major error are reported in Table 4.

The fosfomycin susceptibility rates provided by AD for the collections tested by BD Phoenix and MicroScan WalkAway Plus were 43% and 54%, respectively. BD Phoenix exhibited 6.9% false-resistant results, 69% CA, and modest concordance (κ = 41% IC95% 24–59%). On the other hand, MicroScan WalkAway Plus showed 3.7% of false-resistant results, 72% CA, and modest concordance (κ = 41% IC95% 23–60%). Nevertheless, both automated systems showed poor detection of resistant isolates, with 49.1% and 56.2% false-susceptible results for BD Phoenix and Microscan WalkAway Plus, respectively. Both methods showed a strong deviation of bias towards a single direction (−77% for BD Phoeniex and −52% for WalkAway).

For *E. coli* (*n* = 46) and *K. pneumoniae* (*n* = 50)*,* BD Phoenix registered poor CA (82.6% and 58%, respectively), high VME rates (53.8% and 50%, respectively), and high ME (6.1% and 10%, respectively). It seemed to best perform with *E. coli* compared to *K. pneumoniae*, with 38 vs. 29 isolates correctly categorized, respectively. It also exhibited a relevant number of false-resistant results (*E. coli n* = 2 of 33, *K. pneumoniae n* = 1 of 10) and underestimated MIC values, with 6 and 20 VME for *E. coli* and *K. pneumoniae,* respectively. *E. coli* was the only species with more than 40 isolates among MicroScan WalkAway Plus tested strains. A suitable CA (90.4%) and ME rate (0%) with an unacceptable VME (57.1%) were shown.

## 3. Discussion

The increasing prevalence of MDR *Enterobacterales* spp. represents a challenge for the treatment of infection and new molecules are needed. However, the old fosfomycin may be a choice for the treatment of MDR bacteria. The AST result influences the choice of antibiotic treatment. For this reason, the reliability of AST is the objective of clinical microbiologist. 

The reference method for fosfomycin susceptibility testing is a laborious and time-consuming test, often incompatible with clinical laboratory routine.

Several commercial ASTs are available for Fosfomycin; however, with low agreement with reference agar dilution [15,16,17,18].

This study evaluated the performance of two automated systems largely used in lab routine for the determination of fosfomycin susceptibility in EB clinical isolates. Fosfomycin MICs were determined by AD as the reference method. Both systems provided low CA rates and high percentages of false-resistant and false-sensitive results.

BD Phoenix exhibited poor correlation with AD showing a low CA rate (69%) and a relevant categorical discrepancy compared to reference method with 49.1% of VME rate. These findings are consistent with a previous study that showed high VME rates for ESBLp *K. pneumoniae* and ESBLp *E. coli* (12% and 12.5%, respectively) [16], suggesting fosfomycin MIC underestimation of BD Phoenix in comparison to the reference method. The strong bias deviation (−77%) observed in this opinion is consistent exsisting with literature data [17].

Similarly, an unacceptable VME rate (56.2%) was achieved with Microscan WalkAway Plus suggesting fosfomycin MIC underestimation, with unreliable susceptibility results, confirmed by bias calculation (−52%). These findings are inconsistent with previously reported data for *Pseudomonas aeruginosa* [19,20]. The different micro-organism tested could explain the discrepancy of results.

The two false-resistant results concerned two strains of *Proteus mirabilis*. It is necessary to conduct further investigations to understand the relationship between bacterial species and fosfomycin MIC overestimation achieved by Microscan WalkAway Plus.

Despite BD Phoenix achieving a 10% of ME, we noted that all EB isolates with MIC > 64 mg/L were correctly categorized, suggesting a reliable result in case of elevated MIC value.

Separate analysis according to most frequent EB species for BD Phoenix showed low accuracy and high categorical agreement discrepancy in *K. pneumoniae* and *E. coli.* These findings contrast with a study by Aprile and coll., which reported better performance of BD Phoenix for *K. pneumoniae* KPC compared to *E. coli* KPC, with high agreement between AD and BD Phoenix results (CA > 94% and VME < 2% for both microorganisms) [21]. We speculate that the difference could be explained with the low level of FF-R rate reported. Reliability of *E. coli* AST with Miscoscan WalkAway Plus was calculated. The results confirm the trend of fosfomycin MIC underestimation. No data are available in the literature to support our findings.

The FF AST is a challenge for clinical microbiologists. Existing literature shows a poor concordance between commercial tests and reference AD method. The gradient test displays variable CA (82–92.5%) and VME (2.1–70%) for *Enterobacterales* spp. [22,23,24,25]. We suggest that the bacteria species tested could explain the high variability. An acceptable CA (100%) and none ME and VME for *E. coli* is reported. On the other hand, the AST for *K. pneumoniae* seems to be more unreliable, especially in correctly classifying FF-resistant strains (52–82% of CA, 2.1–78.8% of VME and 5.8–76.7% of ME) [15,25,26,27]. Automated systems provide unreliable results compared to the reference method, with a low capacity to identify susceptible strains [15,16,17].

The difference between AD and broth-based AST inoculum can explain the poor agreement with reference method [25] or by difference diffusion of FF in the medium [17].

Limited data are available for Gram-positive bacteria. EUCAST defines clinical breakpoints only for *Staphylococcus* spp. The CLSI provides BPs for *Staphylococcus* spp. and *Enterococcus* spp. The FF AST broth-based method is unreliable for *S. aureus*, although the Etest method shows better performance (CA = 84.1%, EA = 98.7, ME = 1.3) [14,27].

Recently, commercial AD is available for application in clinical microbiology laboratory routine. An acceptable value of CA, no VME, and ME for both for Gram-negative and Gram-positive bacteria compared to the AD method. These data suggested that commercial method based on agar dilution can be used to replace home-made and laborious AD [27,28].

Our findings are in accordance with data in the literature [15,16,17,18,22,23,24,25] and highlight the low agreement between commercial test and reference method. We suggest that it could be reported “resistant” if tested with MicroScan WalkAway Plus (MIC > 32 mg/L). On the other hand, a result of fosfomycin MIC > 64 mg/L of BD Phoenix is reliable. Susceptibility result must be confirmed with reference method, especially if fosfomycin is used during therapy.

In conclusion, both automated systems provided unacceptable rates of MEs and VMEs, indicating that determination of fosfomycin susceptibility is unreliable, with frequent underestimation of MIC values.

Limitations of the present study include the small number of EB isolates collected in a single center and the use of two different collections of strains. The high number of Fosfomycin-resistant EB isolates represents a notable strength.

Further studies are needed to confirm and expand our findings.

## 4. Materials and Methods

### 4.1. Study Design

We compared the result of fosfomycin susceptibility test of two different collections of *Enterobacterales* spp. using the reference AD method. Each collection consisted of 100 non-duplicate clinical EB collected from several clinical specimen, including urine (*n* = 105), blood (*n* = 20), bile (*n* = 19), wound (*n* = 18), rectal swab (*n* = 17), peritoneal fluid (*n* = 11), bronco-lavage (*n* = 6), joint fluid (*n* = 1), and tissue biopsy (*n* = 3).

### 4.2. Bacteria Isolates and Detection of MDR Strains

Species identification for each strain was performed using MALDI-ToF technology (MALDI Biotyper Systems, Bruker Daltonics, Bremen, Germany). Phenotypic characterization of MDR strains was performed using the Total ESBL Confirm Kit (Rosco, Taastrup, Denmark) to identify ESBL production in the case of cefotaxime (CTX) and/or ceftazidime (CAZ) minimal inhibitory concentrations (MICs) > 1 mg/L. MASTDISCS combi Carba plus disc system (Mast Group Ltd., Bootle, UK) was used to characterize carbapenemase producers when meropenem (MEM) MIC was >0.125 mg/L. Detection of carbapenemases genes was carried out using the Xpert Carba-R assay (Cepheid, Sunnyvale, CA, USA).

### 4.3. Reference Method (Agar Dilution Method)

The reference minimum inhibitory concentration for fosfomycin was determined by AD according to international guidelines [29,30] with in-house Mueller Hinton II Agar (Becton Dickinson, Franklin Lakes, NJ, USA) containing fosfomycin (Nordic Pharma, Gentilly, France), with two-fold dilutions from 16 mg/L to 64 mg/L, and 25 mg/L of Glucose-6-Phosphate (Sigma, Germany), as previously described [16]. The addition of Glucose-6-Phosphate to agar is mandatory because the drug needs glucose-6-phosphate transporters to penetrate into the cell [3]. Each isolate, after 24 h of grown on blood agar (Becton Dickinson, MD, USA), was tested in technical triplicate on separate days. In brief, 0.5 McFarland suspension was performed and spotted on Mueller Hinton II Agar to final inocolum of 1 × 10^4^ CFU/spot. The plates were incubated at 35 ± 2 °C for 16–20 h. The lowest concentration of fosfomycin that inhibited bacterial growth in at least two out of three spots was defined as isolate MIC. EUCAST breakpoints (Version 13.0) for fosfomycin *iv* was applied to define susceptibility (susceptible ≤ 32 mg/L, resistant > 32 mg/L).

### 4.4. Commercial Method

The determination of fosfomycin MIC with BD Phoenix (Becton Dickinson, MD, USA) and MicroScan WalkAway Plus (Beckman Coulter, CA, USA) was performed according to the manufacturer’s instructions with the test panels NMIC/ID-94 and Neg Combo-83, respectively. ASTs were set up starting from isolates grown 24 h on blood agar. BD Phoenix fosfomycin AST is based on three dilution (16 mg/L, 32 mg/L and 64 mg/L) with a MIC range of ≤16 mg/L to >64 mg/L. MicroScan WalkAway panel has one dilution of fosfomyin (32 mg/L). Antibiotic susceptibility was interpreted according to EUCAST breakpoints.

### 4.5. Statistical Analysis

The evaluation of BD Phoenix NMIC/ID-94 panel and MicroScan WalkAway Neg Combo-83 panel was performed by calculating categorical agreement (CA), major error (ME) and very major error (VME) according to the International Organization for Standardization (ISO standard 20776-2), using AD susceptibility results as reference (CA ≥ 90%, VME ≤ 3%) [30]. Briefly, CA was defined as the proportion of isolates classified in the same susceptibility category by reference method and the methods under evaluation. VME and ME were defined as variations in interpretation from resistant to susceptible (false susceptible) and from susceptible to resistant (false resistant) compared to reference method, respectively. The bias of the method was calculated as indicated by the new ISO 20776-2:2021 and it is defined as the deviation of the test compared to reference method [31]. The concordance agreement was assessed using κ-Cohen [32].

Finally, automated systems performance for bacterial species represented by more than 40 isolates was also assessed.

## Figures and Tables

**Table 1 antibiotics-12-01106-t001:** Multi-drugs resistant *Enterobacterales* isolates tested for fosfomycin susceptibility using BD Phoenix and Microscan WalkAway Plus.

AST Method/Species		ESBLp-EB	Cp-EB
**BD Phoenix**		33 (33)	17 (17)
	*Escherichia coli*	41 (19)	2 (1)
	*Klebisella pneumoniae*	28 (14)	28 (14)
	*Enterobacter cloacae*	-	50 (2)
**MiscroScan WalkAway Plus**		30 (30)	16 (16)
	*Escherichia coli*	50 (21)	7.2 (3)
	*Klebsiella pneumoniae*	28 (7)	40 (10)
	*Enterobacter cloacae*	-	12 (2)
	*Proteus mirabilis*	2 (2)	-
	*Morganella morganii*	-	-
	*Citrobacter koseri*	-	-

All data are shown as relative (%) and absolute (*n*) frequencies, unless otherwise stated. Abbreviations: AST: antimicrobial susceptibility testing; ESBLp-EB: extended-spectrum β-lactamase-producing *Enterobacterales*; Cp-EB: carbapenemase-producing *Enterobacterales*.

**Table 2 antibiotics-12-01106-t002:** Comparison between results of BD Phoenix and Agar dilution.

Agar Dilution				
MIC (mg/L)	≤16	32	64	>64
≤16	32	5	8	13
32	1	2	3	4
64	2	1	1	5
>64	0	0	0	23

**Table 3 antibiotics-12-01106-t003:** Comparison between results of MicroScan WalkAway and Agar dilution.

Agar Dilution				
MIC (mg/L)	≤16	32	64	>64
≤32	49	3	4	22
>32	1	1	0	20

**Table 4 antibiotics-12-01106-t004:** Evaluation of AST method compared to Agar dilution.

AST Method/Species	CA	VME	ME
**BD Phoenix**	69 (69)	49.1 (28)	6.9 (3)
*Escherichia coli* (46)	82.6 (38)	53.8 (6)	6.1 (2)
*Klebsiella pneumoniae* (50)	58 (29)	50 (20)	10 (1)
**Microscan WalkAway Plus**	72 (72)	56.2 (26)	3.7 (2)
*Escherichia coli* (42)	90.4 (38)	57.1 (4)	-

All data are shown as relative (%) and absolute (n) frequencies, unless otherwise stated. Abbreviations: AST: antimicrobial susceptibility testing; CA: categorical agreement; VME: very major error; ME: major error.

## Data Availability

The dataset analyzed during the current study is available from the corresponding author upon reasonable request.

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
