# Peer review of "Performance Evaluation of BD Phoenix and MicroScan WalkAway Plus for Determination of Fosfomycin Susceptibility in Enterobacterales"

_antibiotics, 2023, doi:10.3390/antibiotics12071106_

Round 1

Reviewer 1 Report

This manuscript revealed a bias in the results of clinical antibiotic resistance investigation methods. The manuscript used some simple research methods, but it addresseed a very important problem. It is of high value to correctly understand drug resistance and evaluate the risk of drug resistance. But the tables in this manuscript are too rough and confusing.

Author Response

I would like to thank the reviewer for considerations and suggestions.

The tables in the manuscript were revised.  

Reviewer 2 Report

I have read with interest the manuscript submitted by Bondi et al., since AMR represents a significant concern.

I have just a few minor comments:

Please consider fragmenting some phrases in order to make the content more clear. As an Italian speaker (non-native) I can understand the meaning of some phrases by translating them into Italian - the expressions used make much more sense that way. Maybe consider having someone English-native proofread the manuscript. 

If you utilize more elaborate statistical methods, the significance of your findings would be much higher, considering the low number of cases included.

Row 172 - Our findings are in accordance with data in the literature - please provide references

Please consider fragmenting some phrases in order to make the content more clear. As an Italian speaker (non-native) I can understand the meaning of some phrases by translating them into Italian - the expressions used make much more sense that way. Maybe consider having someone English-native proofread the manuscript. 

Author Response

I would like to thank the reviewer for considerations and language suggestions. We have changed and fragmented some phrases. 

The international guidelines for evaluation of AST vs gold standard suggest to use categorical agreement (CA), essential agreement (EA), minor error (MiE), major error (ME) and very major error (VME) and bias calculation. We follow these criteria. However, with our data we cannot calculate  EA and MiE. We have added a k-cohen value to underline the degree of accuracy and reliability.

Reviewer 3 Report

Bondi et al. “Performance evaluation…” refers to the important issue which is assessment of susceptibility to fosfomycin using automatic methods, however it needs major changes. The Authors assessed susceptibility to fosfomycin on two different analyzers, but for different species. There is no information which strains were analysed on two analyseres. It’s difficult to compare these results and the purpose isn’ t clear. It should be changed. A big limitation in this study, about the Authors write, is the small number of tested strains e.g. 2 E. cloacae, 3 M. morgannii, 3 C. freundii etc. Drawing conclusions on this basis is inappropriate.

Author Response

I would like to thank the reviewer for the considerations.

We reported the information about species tested with two system at line 61-64.

The goal of this study is to evaluate the result of fosfomycin AST of automated methods. For this porpouse we used two collections of Enterobacterales spp. The collections are different and not comparable to each other. The comparision is made between BD Pheoniex vs Agar dilution and Microscan WalkAway vs Agar dilution, in ordert to assessment the fosfomycin AST results. 

We agree with the reviewer that the small number of isolates is a study limitation. However, no conclusion is drawn about specie with a low number of isolates (ie E.cloacae, M.morganii, C.koserii). We underline the low agreement of commercial methods results for Enterobacterales spp and no for single species (with the exception of E.coli and K.pneumoniae). 

Round 2

Reviewer 1 Report

I think it is acceptable to publish this manuscript.

Author Response

We thank the reviewer for the work done in the review process

Reviewer 3 Report

I suggest to write in the limitations of the manuscript about comparing two different collections on different analyzers.

In line 122 is "antimibitch". I don't know what the Authors meant.

Author Response

We thank the reviewer. 

The suggestions have been added to the manuscript.

 "antimibitch" in line 122 is a typo. The word has been corrected.